# Urinary markers of oxidative stress respond to infection and late-life in wild chimpanzees

**Nicole Thompson González**[1,2]*, **Emily Otali**[3], **Zarin Machanda**[4,3], **Martin N. Muller**[1,3], **Richard Wrangham**[5,3], **Melissa Emery Thompson**[1,3]

**1** University of New Mexico, Department of Anthropology, Albuquerque, NM, United States of America, **2** University of New Mexico, Academic Science Education and Research Training Program, Health Sciences Center, Albuquerque, NM, United States of America, **3** Kibale Chimpanzee Project, Fort Portal, Uganda, **4** Tufts University, Department of Anthropology, Medford, MA, United States of America, **5** Harvard University, Department of Human Evolutionary Biology, Cambridge, MA, United States of America

* gavago@gmail.com

**Data Availability Statement:** The data are currently available publicly on my GitHub page: https://github.com/Gavago/Oxidative-stress-chimpanzee.git

## Abstract

Oxidative stress (OS) plays a marked role in aging and results from a variety of stressors, making it a powerful measure of health and a way to examine costs associated with life history investments within and across species. However, few urinary OS markers have been examined under field conditions, particularly in primates, and their utility to non-invasively monitor the costs of acute stressors versus the long-term damage associated with aging is poorly understood. In this study, we examined variation in 5 urinary markers of oxidative damage and protection under 5 validation paradigms for 37 wild, chimpanzees living in the Kibale National Park, Uganda. We used 924 urine samples to examine responses to acute immune challenge (respiratory illness or severe wounding), as well as mixed-longitudinal and intra-individual variation with age. DNA damage (8-OHdG) correlated positively with all other markers of damage (F-isoprostanes, MDA-TBARS, and neopterin) but did not correlate with protection (total antioxidant capacity). Within individuals, all markers of damage responded to at least one if not both types of acute infection. While OS is expected to increase with age, this was not generally true in chimpanzees. However, significant changes in oxidative damage were detected within past-prime individuals and those close to death. Our results indicate that OS can be measured using field-collected urine and integrates short- and long-term aspects of health. They further suggest that more data are needed from long-lived, wild animals to illuminate if common age-related increases in inflammation and OS damage are typical or recently aberrant in humans.

## Introduction

As aerobic life forms use oxygen for energy metabolism, immune defense, and cell signaling, they are subject to cellular damage from byproducts of oxygen use, i.e. reactive oxygen species [1–3]. The body balances reactive oxygen species with antioxidants and repair mechanisms, yet acute conditions can disrupt the balance to favor reactive oxidants, leading to a state of oxidative stress (OS, Finkel & Holbrook [4–6]). As proximate damage accumulates with time, it

**Funding:** This work was funded by the National Institute of Aging and the NIH Office for Research on Women's Health award R01-049395 (MET, MM, ZM, EO, RW, https://www.nia.nih.gov/), and by the Leakey Foundation (https://leakeyfoundation.org/), the National Science Foundation (BCS-1355014, BCS-0849380, MET, MM, ZM, EO, RW, https://www.nsf.gov/), the University of New Mexico, and Harvard University, the Foundation for the National Institutes of Health (K12 GM088021, NTG), National Science Foundation (NCS-FO-1926352/1926737 to MET, ZM, MM. The funders had no role in study design, data collection and analysis, decision to publish, or preparation of the manuscript.

**Competing interests:** The authors have declared that no competing interests exist.

can hinder physiological function, thus several theories place OS as a central mechanism of biological aging and age-related disease [4–6]. OS markers have been examined in relation to aging and challenges among animals living in the wild (particularly malondialdehyde as thiobarbituric acid reactive substances i.e. MDA-TBARS, reactive oxygen metabolites, and various antioxidants, [7]) and in the laboratory [8], usually in blood or other cells. However, several measures of OS are possible in urine, suggesting it may be feasible to develop a non-invasive toolkit to monitor OS in field settings where blood sampling is not possible. The objective of this paper is to examine a suite of urinary OS markers to determine their suitability for monitoring both acute, short-term health challenges (e.g. infections and wounding) as well as long-term patterns of aging within a population of wild primates.

Organisms are expected to experience fundamental tradeoffs between life history functions such that, for example, energetic investments in immediate survival (e.g., immunity) and reproduction detract from longevity. Such marked energetic investments often result in a state of OS, which over a lifetime can lead to cumulative damage and functional loss, i.e. aging [7,9–12]. Cumulative damage is particularly likely as endogenous antioxidants and the function of DNA repair mechanisms decline with age [13–17]. For this reason, oxidative stress has been posited as a mechanism of aging itself over half a century [4,11,18,19]. Nevertheless, more recent comparative research reveals that oxidative damage might actually decline with age in adults of some species (e.g. naked mole rats, Andziak & Buffenstein [20] wild boar, Gassó et al., [21]; rhesus macaques, Georgiev et al., [22]), and antioxidant activity increases with age and reproductive investment (humans, Bolzán et al., [23]; house mice, Garratt et al., [24]; [21]; shrews, Hindle et al., [25]; collared flycatchers, Markó et al., [26]; voles, Ołdakowski et al., [27]). Therefore, the empirical evidence indicates that the role of oxidative stress in aging and life history is far from straightforward.

A central cause of oxidative damage is the cellular or inflammatory immune response to infectious disease [4,6]. When exposed to a pathogen, the NADPH complex in phagocytes initiates the production of a cascade of reactive oxygen species such as superoxide and hydrogen peroxide, and pro-oxidant cytokines such as TNF and IL-1 [28–30]. Reactive oxidants damage pathogens but also cause collateral damage to self by oxidizing DNA, proteins, and lipids in tissues and in circulation (see Fig 1 in Selman et al., [10]). The accumulation of oxidative damage in cells and the independent decline in adaptive immunity with age can lead to the inflammatory response's common over-activation in old age [31–33]. In humans, a chronic state of "inflammaging" is characterized by the self-perpetuating relationship of oxidative damage and inflammation in older individuals, and is involved in the pathogenesis of nearly all age-related disease including atherosclerosis, cancer, cataracts, and neurodegenerative disease [32,34,35]. However, recent studies indicate that the experience of chronic inflammation and its pathological consequences vary considerably across human populations [36–39], calling into question whether these diseases are natural and inevitable sequelae of human aging or consequences of modern environments. Evaluating the broader occurrence of chronic inflammation during aging among non-human primates could, therefore, illuminate the evolutionary roots of these common human ailments.

## Markers of oxidative stress

While there is great interest in examining the role of redox status, i.e. the balance between oxidants, antioxidants, and repair, in health and longevity among wild primates, field researchers are currently in need of non-invasive methods to do so. Often, OS markers have been assayed in isolation, because they are simple to quantify (e.g. antioxidants) or inexpensive (e.g. MDA, [7]). However, redox status involves a complex cascade of events, involving protection,

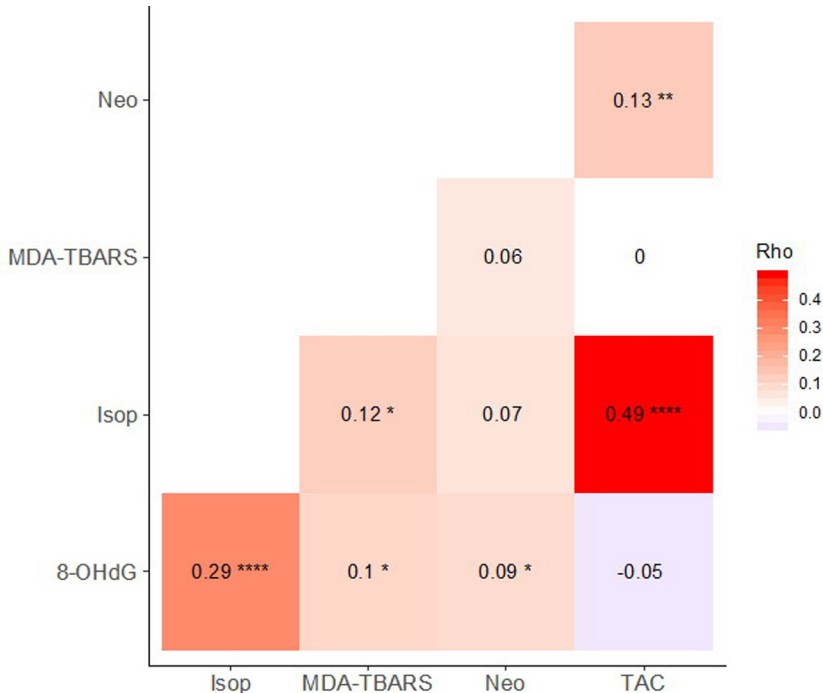

**Fig 1. Correlations between OS biomarkers.** Spearman's rho written in heat tiles, with significant correlations indicated as **** p < 0.0001, *** p < 0.001, ** p < 0.01, * p < 0.05, and † p < 0.10.

damage, and repair of multiple target molecules at multiple time points and in multiple bodily regions [40,41]. Many sources therefore recommend measuring redox status via multiple markers that each provide an index of particular processes and responses [7,12,42]. In this study, we selected several candidate markers for which commercial urinary assays are available, that are known to be stable, and that have been in common use in clinical studies with evidence of response under stress conditions (e.g. disease).

Neopterin is a marker of the inflammatory immune response in primates and consequently an indicator of oxidative stress, as it is produced along with reactive oxygen species by activated macrophages [43]. Its stability and response to viral and bacterial infection has been well-documented in laboratory [44,45], zoo [46], and field-based studies of primates [47]. Further, and consistent with patterns of immunosenescence and inflammaging, neopterin increases with age in humans [48,49] and Barbary macaques [50].

Products of oxidative damage to DNA, lipids, and proteins are also excreted in urine. Many such products have been validated as markers of acute stressors and are seen to correlate with aging in human and laboratory animal models [51]. 8-hydroxy-2'-deoxyguanosine (8-OHdG) is a product of the oxidation of the nucleotide guanine that is excreted in urine after base excision repair [52]. 8-OHdG increases in response to toxin exposures (CCl4, Kadiiska et al., [53]) and is higher among individuals with vs. without various chronic diseases [52,54]. DNA damage increases with age in several critical tissues in both humans and rodents [14,16,55–58].

Popular measures of the oxidation of lipids (a.k.a. lipid peroxidation), and corresponding damage to cellular membranes, include malondialdehyde (MDA) and, more recently, isoprostanes [51]. While MDA has been more commonly assayed across taxa due to its low cost and ease of measurement, it is nonspecific to lipid peroxidation and may conflate independent physiological processes (e.g. dietary MDA content, blood oxygen tension, Il'yasova et al.,

[51,59]. Isoprostanes, which are prostaglandin-like compounds that result from interaction of reactive oxygen species with arachidonic acid, appear more reliable and are growing in popularity [51,60–63]. Both markers increase in response to toxin exposure in rodents and are elevated in humans during various chronic diseases and smoking [64–68]. Humans and rodent models show increases in lipid peroxidation with age (MDA-TBARS, İnal et al., [69]; isoprostanes, Praticò, [70]); however, field studies in rhesus macaques and Soay sheep show no variation in MDA with age [22,71].

OS can also be assessed via protein oxidation products, but these are less stable than other markers. To date, established markers such as protein carbonyls and dityrosine are difficult to detect in urine in a consistent and reliable way [51,53,72].

Protection from oxidative damage can also be measured non-invasively, as urine contains viable antioxidants from both endogenous and dietary origin. Total antioxidant capacity (TAC) is a measure of oxidative resistance indicated by the ability of a medium to prevent a redox reaction between a controlled reactant and reagent [73] and has been validated to negatively correlate with isoprostane levels [74]. Endogenous antioxidants, in particular, often vary with age in a complex manner, as their basal expression increases but their expression in response to external stimuli decreases with age in humans and rodents [75].

## Study system

We evaluated markers of OS in a mixed-longitudinal sample among wild chimpanzees in the Kanyawara community in the Kibale National Park, Uganda. Chimpanzees are long-lived (maximum lifespan > 60 years) and share a close evolutionary relationship with humans. Data from longevous animals are in need to clarify the role of OS in shaping life history trade-offs and aging [12,42], and data on chimpanzees provide a particularly essential comparison to evaluate evolutionary changes in the aging process during human evolution. While experimental studies of OS with controlled exposure to stressors have yielded important insights, naturalistic studies are necessary to examine the arc of senescence against the realistic evolutionary backdrop of limited resource abundance and heavy antigenic exposure [7,10,76,77]. Our study of the Kanyawara community also has the distinct advantage of dense, long-term urinary sampling, with daily observations of individual health status. This dataset provides an opportunity for analyzing responses to natural experiments (e.g. infection, injury) and aging within individuals, and so reduces the potential confound of mortality selection to which cross-sectional designs are prone [10,12,22,71,78,79].

This study aims to provide a biological validation of a panel of OS markers derived from field-collected urine. While products of OS are broadly applicable across animals, and urinary assays for OS are in common use for humans, we wanted to evaluate if these markers were sensitive enough that we could detect responses to major immunological challenges, even under conditions of opportunistic sample collection and under field collection and storage constraints. We also wanted to evaluate whether different markers yielded different information. Our biological validation had three facets. First, while the various markers we used reflect different aspects of redox status, we expected them to be moderately correlated with one another. Specifically, we expected markers associated with infection (neopterin) and oxidative damage (8-OHdG, F2-isoprostanes, MDA-TBARS) to be positively correlated with one another, and negatively correlated with antioxidant capacity (TAC). Second, we expected that these markers should exhibit a significant acute response to active immunological challenges: an epidemic of respiratory illness and cases of severe wounding. We expected levels of neopterin and markers of oxidative damage to increase in response to these challenges, and for TAC to decrease as antioxidant defenses were exhausted. These tests served as a proof of concept that these

markers could reliably detect acute insults, and further to discriminate whether the different challenges (wounds vs. respiratory viral infection) yielded different oxidative responses. Our third major goal was to extend these findings to test the hypothesis that the cumulative immunological burden of living under pathogen exposure leads to an increase in oxidative damage with age. Here, we examined the correlation of OS markers with age using a mixed-longitudinal sample comprised of repeated samples of 36 adult individuals. Because this approach introduces the possibility of mortality selection, we additionally looked at how OS markers changed in the several years leading up to death in a more densely sampled longitudinal dataset from 4 individuals. We expected markers of oxidative damage to increase, and antioxidant capacity to decrease, both as chimpanzees aged and as death approached.

## Methods

### Study site and data collection

Research permissions for the study were granted by the Uganda Wildlife Authority, Uganda National Council for Science and Technology, and Makerere University Biological Field Station. Protocols were approved by the Institutional Animal Care and Use Committees of Harvard University and the University of New Mexico.

Data were collected between 2008 and 2017 on the Kanyawara community of wild, unprovisioned chimpanzees in Kibale National Forest, Uganda. Ages of most subjects born after 1987 were known to within one month based on long-term demographic records collected during near-daily observations. Female chimpanzees often exhibit primary dispersal at sexual maturity, and thus new nulliparous females were assigned an age at immigration of 13 years, based upon the average age of females who have dispersed from the community. Ages of older individuals, most of whom were first identified in 1983, were estimated based on body size, if immature at first sighting, or on signs of relative age, including condition of body and hair and the presence of dependent offspring (see Hill et al., [80,81]).

Trained staff of the Kibale Chimpanzee Project recorded daily clinical signs of all observed individuals, including coughing, sneezing, wounds, limping, and skin conditions (see Emery Thompson et al., [82]). Coughs were further noted as dry or wet/productive, while wound documentation included descriptions of location, severity, and healing process. Observers collected urine samples opportunistically from identified individuals, using a disposable pipette to transfer urine from a plastic bag or leaf into 2.0 ml tubes. Urine samples had minimal dirt contamination and no contamination with feces, were stored immediately on ice, and were frozen within 0–14 hrs. OS marker concentrations exist in a more stable state in urine than in other media [51], and most have been shown to vary minimally after remaining at room temperature upon collection for up to 16–24 hours (urinary neopterin, Heistermann & Higham, [83]; plasma TAC, Koracevic et al., [84]; urinary MDA, Lee & Kang, [85]; urinary 8-OHdG, Matsumoto et al., [86]). While no formal study has assessed the stability of urinary isoprostanes upon collection, concentrations in thawed urine samples decrease <15% after storage in the dark at room temperature for 2 days [87]. Chimpanzee samples were transported frozen and on ice to the Comparative Human and Primate Physiology Center at UNM and stored at < -30°C until further analysis. Using linear regression, we found a weak negative relationship between storage time and MDA-TBARS concentration (ß = -0.3, p = 0.046), but none in any other biomarker (8-OHdG ß = 0.00008, p = 0.56; isoprostanes ß = -0.13, p = 0.06; neopterin ß = -0.02, p = 0.13; TAC ß 0.02, p = 0.67). We therefore reran age-related analyses of MDA-TBARS using samples stored for ≤ 3 years total (mixed-longitudinal), and with a difference in storage time ≤ 3 years (longitudinal, lead up to death) and found no change to our original results.

## Assays for biomarkers of oxidative stress

Samples were assayed for 5 different urinary OS biomarkers using kits designed for urine. We used ELISAs for 8-hydroxy-2'-deoxyguanosine (**8-OHdG**, Catalog No. KOG-200S/E Genox JaICA), 15-Isoprostane $F_{2t}$ (**isoprostanes**, Prod. No. E85 Oxford Biomedical Research), and **neopterin (**Ref. RE59321, IBL International), and colorimetric assays for malondialdehyde as thiobarbituric acid reactive substances (**MDA-TBARS**, Item No. 700870 Cayman Chemical) and total antioxidant capacity (**TAC**, ABTS$^+$ method, Item No. 709001 Cayman Chemical). In all cases, assays were performed according to the manufacturers' suggested protocols and dilutions, as chimpanzee samples yielded values in the expected ranges for those assays. For isoprostranes and MDA-TBARS, protocols included administering an enhanced dilution buffer and precipitation agent to samples to increase assay specificity. We measured specific gravity of all samples with an Atago$^{TM}$ handheld refractometer (PAL-10S) and standardized biomarker concentrations for specific gravity following [88]. These corrections can lead to conspicuous outliers in very dilute samples. After visually inspecting the data for inflated biomarker values at low dilutions, we omitted isoprostanes, neopterin, and TAC values for samples with SG < 1.004. Average recovery of enzyme immunoassays on spiked urine samples was 116% for 8-OHdG (88–137%) and 114% for isoprostanes (106–123%). Behringer et al. [46] report 100.3% recovery of neopterin from spiked chimpanzee urine samples. The proportion of samples with assayed values above kits' minimum concentration of detection were 99.7% for 8-OHdG, 100% for isoprostanes, 98.6% for MDA-TBARS, 99.7% neopterin, and 99.9% for TAC. Intra-assay CVs were 5.9% for 8-OHdG, 5.8% for isoprostanes, 3.1% for MDA-TBARS, 5.9% for neopterin, and 2.8% for TAC. Inter-assay CVs for high and low controls were 12.2% and 13.6% for 8-OHdG, 10.1% and 14.2% for isoprostanes, 3.7% and 6.7% for MDA-TBARS, 7.9% and 10.3% for neopterin, and 11.7% and 13.6% for TAC.

We aimed to select samples that had never previously been thawed, but this was not always possible as samples are periodically used for other laboratory studies. Sample values were highly correlated after first and second thaws within all assays (range Spearman $r_S$ = 0.94–0.99). We examined changes in biomarker concentrations between thaws by calculating the average proportional change in values between first and second thaws and performing a 2-tailed Wilcoxon rank sum test for significant differences: 131% for 8-OHdG (n = 36, W = 504, p = 0.11), 116% for isoprostanes (n = 36, W = 549, p = 0.27), 86.7% for MDA-TBARS (n = 37, W = 792, p = 0.26), 99.8% for neopterin (n = 30, W = 468, p = 0.80), and 130% for TAC (n = 35, W = 285, p < 0.001). We therefore aimed to assay samples on their first thaw for 8-OHdG, isoprostanes, and TAC. To increase sample size for underrepresented individuals, we assayed some samples that were previously thawed once (1–4 hours). The total proportion of samples assayed after their first thaw was 89% for 8-OHdG, 67% for isoprostanes, and 78% for TAC. MDA-TBARS and neopterin were assayed on either the first thaw or the second, after having been previously thawed for <1 hr. We were unable to assay all samples for all markers due to small sample volumes, thus the sample sizes vary by marker.

## Validation approach, subjects, and sample selection

We conducted 4 different types of validation analyses to examine the relationship of oxidative stress biomarkers with aging, infection, and injury, as below. We assayed samples of individuals $\geq$ 10 years old, except in one case study of a 6 yr old male (BT) that experienced a serious injury.

1. <u>Before, during, and after respiratory epidemic:</u> Viral respiratory infections reliably induce macrophage responses and oxidative stress in humans [29, 89]. A severe respiratory

epidemic broke out in the Kanyawara community in 2013, during which 87% of chimpanzees were observed to have wet, productive coughs. Respiratory epidemics have been a major source of mortality at Kanyawara, and in the 2013 outbreak, traced to human rhinovirus C, five individuals died (~ 9%, [82, 90]). We used 265 samples from 27 individuals (12 M, 15 F) to examine within-individual changes in OS biomarkers between periods before, during, and after the 2013 epidemic. For "during" the epidemic, we chose samples from the 4-month period when the weekly proportion of community members showing signs of coughing and sneezing were elevated (date ranges in Table 2). This period was bookended by severe peaks in respiratory signs. We then identified 7-month periods "before" and "after" the prolonged outbreak. Given a single week when several members showed severe respiratory signs 2 months prior to the prolonged outbreak, we excluded these 2 months and restricted our control sample to the window 2–7 months before the outbreak. All individuals were sampled for at least two periods for any given biomarker. For specific sampling per biomarker, see S1 Table.

2. Before, during, and after serious injury: As a further proof-of-concept that acute challenges lead to OS, we examined OS biomarker responses to traumatic injury in case studies of three individuals. We selected cases that were severe enough that they posed a strong risk of bacterial infection and that did not overlap with the respiratory epidemic. Two chimpanzees, GG (14 yo F) and BT (6.3 yo M), were caught by wire hunting snares, which pose significant risk of infection because the wire embeds in the flesh [91]. While veterinary intervention is sometimes possible, it was not in these cases, thus neither individual received any antibiotics or other treatment. For each individual, the injuries resulted in the loss of 4 fingers. The third case, PB (18 yo M), was attacked and severely wounded by chimpanzees from a neighboring community (such attacks are often fatal). He received bite and/or tear wounds on both thighs, chest, face, testes, anus, heel, and hand; these resulted in significantly impaired movement and took >2 weeks to heal. All subjects were sampled over the 3 months before the injury, during the period between injury and healing (by visual assessment for bleeding, inflammation, infection, and scabbing), and for 3 months after the injury healed. For sampling date ranges by individual and biomarker, see S2 Table.

3. Quarterly mixed-longitudinal: To assess natural variation by age and sex in OS biomarkers, we assayed 594 samples from 36 chimpanzees (18 F, 18 M) aged 10+ years over a period of approximately 5 years (range 0.75–7.8 yrs/individual) for an average of 16.5 ± 10.2 samples/individual. We attempted to represent subjects consistently over time by selecting approximately one sample in each quarter-year. Mixed-longitudinal samples did not include samples collected during and after the respiratory epidemic or severe injury. To evaluate if individuals experience more pronounced oxidative damage simultaneously with physical decline, we further examined age-related changes in biomarker concentrations in past-prime individuals alone, i.e. > 35 years old, as lean body mass begins to decrease and mortality rates increase near this age (Emery Thompson et al., *in press*). This subset consisted of 9 individuals with 140 samples, or 16.5 ± 10.2 samples each and collected over an approximate 3.9-year period (range 0.01–7.8 years).

4. Lead up to death: We additionally conducted more intensive longitudinal sampling of 4 individuals who died during the study period. Three were among the oldest chimpanzees known from Kanyawara: OU (F age 48, cause unknown), ST (M age 58, respiratory illness), and BL (F age 56 yrs, cause unknown) while the fourth died suddenly in his prime (KK, M age 28, respiratory illness plus physical trauma). We sampled each individual throughout the 6 to 9 consecutive years preceding death, for an average of 5.0 ± 3.6 samples per

individual-year (total n = 161 samples). Because KK's relative young age at death, we ran statistical models both including and excluding him. For sampling date ranges per individual and sample numbers per year see S3 Table.

## Statistical analysis

Oxidative stress varies throughout the day, as a result of the metabolic costs of daily activities and circadian rhythms in antioxidant production [92,93]. We attempted to minimize time of day effects by prioritizing samples taken in the morning hours; however, this was not always possible. We therefore assessed the relationship between time of sample collection and bio-marker concentrations using a generalized linear mixed model (GLMM, in R packages "lme4" v. 1.1–21 and "lmerTest" v. 3.1–1) with a gamma error distribution and log-link function. Models included biomarker concentration as a response, hour of sample collection as a fixed effect, and individual as a random intercept with hour of collection as a random slope. To minimize the influence of potential outliers, these models were applied to samples collected during baseline conditions, i.e. not collected during or after a respiratory epidemic or injury, and not from the four individuals whose deaths were the focus of our analyses above. Because MDA-T-BARS, neopterin, and TAC varied significantly with time of day (S4 Table), we thereafter expressed their concentrations as time corrected values, calculated as residuals from predicted values, calculated using the population slope and intercept of baseline models.

As individuals were occasionally sampled more than once on a given day, our final dataset used average daily values of either unadjusted biomarker concentrations (8-OHdG and iso-prostanes) or their time-adjusted residuals (MDA-TBARS, neopterin, and TAC). All Ns reported reflect the statistical sample size = days of sampling.

We constructed linear and generalized linear mixed models (LMMs and GLMMs) to test the predictions of our validation hypotheses. For all models, we assessed the appropriateness of generalized error distributions using the "descdist" function (R package "fitdistrplus" v. 1.0–14) prior to modeling and visually inspecting quantile-quantile plots of model residuals. To assess correlation between markers, we examined pairs of biomarker concentrations as fixed effect and response, with a random effect of individual. In Table 1, we outline the structures of validation models to assess OS marker variation by infection and age. All data preparation and analyses were performed using R [94].

While it may in the future be desirable to collapse biomarker values into fewer dimensions or a single index, this was not practical in the given dataset because not all samples had sufficient volume for all markers, and, as our study evolved, newly-added isoprostanes were assessed in different samples to aim for urine not previously thawed. Thus, too few samples contained the full complement for dimension reduction, and we chose to focus on the individual markers in this study.

**Table 1. Structure of linear mixed models of OS biomarkers' response to infection and age.**

| Analysis | Response | Predictors | Random intercept |
|---|---|---|---|
| Before during and after epidemic | Biomarker | Time period + Sex + Age[†] | Chimp ID |
| Before during and after injury | Biomarker | Time period + Sex + Age | Chimp ID |
| Mixed-longitudinal | Biomarker | Sex + Age | Chimp ID |
| | | Sex + Age + Sex*Age | |
| Lead up to death | Biomarker | Years until death[†] + Sex | Chimp ID |

[†]Continuous variables always mean-centered and scaled by their standard deviation

## Results

### i. Correlations between OS biomarkers

Isoprostanes, MDA-TBARS, and neopterin all positively correlated with 8-OHdG (Fig 1 and S5 Table). The strongest correlation of these was between isoprostanes and 8-OHdG, markers of lipid and DNA damage, respectively. Isoprostanes and MDA-TBARS, both measures of lipid peroxidation, positively correlated with one another, however weakly (Spearman's correlation, $r_s$ = 0.12, p = 0.047, 2-tailed). TAC positively correlated with both neopterin and isoprostanes, and did not correlate with 8-OHdG or MDA-TBARS, counter to predictions that TAC would negatively correlate with markers of oxidative damage.

### Validation 1. Before, during, and after respiratory epidemic

Several OS biomarkers changed significantly during and/or after the respiratory outbreak, though not always in the ways predicted. Only 8-OHdG was elevated during the epidemic period compared with before (Table 2 and Fig 2A), and levels of 8-OHdG remained elevated after the epidemic was over. MDA-TBARS (Fig 2C) and neopterin (Fig 2D) did not increase during the epidemic itself, but were increased in the recovery period after the epidemic. Neither isoprostanes nor TAC varied over the course of the epidemic (Fig 2B and 2E). Notably, several samples from the 4 adults who died during the epidemic (1 young infant also died and was not sampled) were high for 8-OHdG, MDA-TBARS, or neopterin before the period when respiratory signs were observed.

### Validation 2. Before, during, and after severe injury

Open wounds provide a different kind of infection risk than respiratory viruses, and accordingly the responses of OS biomarkers were expected to differ. Both isoprostanes and neopterin increased during injury relative to pre-injury levels (Fig 3B and 3D and Table 3). The three subjects showed no significant variation in 8-OHdG, MDA-TBARS, or TAC levels in response to injury (Fig 3A, 3C and 3E).

One individual (GG, snared, Fig 3) exhibited an unusual pattern for all of the biomarkers examined. While GG's isoprostanes increased markedly in response to injury, unlike the other individuals, they continued to increase after the wound was visibly healed. On the other hand, while her levels of neopterin were similarly high during the injury phase as the other two, and her post-healing levels were similarly low, she had already exhibited highly elevated neopterin

**Table 2. Variation in OS biomarkers between periods of before, during, and after a severe respiratory epidemic.** Betas and standard deviations of contrasts from generalized linear mixed effects model and percentage of overall variance explained by individual ID as a random effect. Betas of significant contrasts in bold.

| OS Biomarker | $n_{individuals}$ | $n_{samples}$ | Before → During | Before → After | During → After | Sex (M) | Age | % RE |
|---|---|---|---|---|---|---|---|---|
| 8-OHdG | 26 | 258 | **0.15 ± 0.06** * | **0.23 ± 0.07** *** | 0.07 ± 0.07 | -0.1 ± 0.08 | 0.01 ± 0.04 | 5.65 |
| Isoprostanes | 18 | 133 | *-0.2 ± 0.11* † | -0.08 ± 0.13 | 0.11 ± 0.13 | -0.21 ± 0.21 | -0.04 ± 0.09 | 17.3 |
| MDA-TBARS | 22 | 191 | *-1.66 ± 0.94* † | 0.91 ± 1.08 | **2.57 ± 1.01** * | -0.37 ± 1.25 | **1.45 ± 0.56** * | 12.74 |
| Neopterin | 25 | 229 | 0.16 ± 0.1 | **0.54 ± 0.11** *** | **0.37 ± 0.11** *** | 0.01 ± 0.11 | 0.03 ± 0.05 | 4.42 |
| TAC | 20 | 172 | *-0.89 ± 0.46* † | 0.06 ± 0.53 | *0.95 ± 0.52* † | -0.35 ± 0.62 | -0.14 ± 0.28 | 12.09 |

† p < 0.10,

* p < 0.05,

** p < 0.01,

*** p < 0.001

Sample date ranges for Before: 10/01/2012–2/28/2013; During: 5/3/2013–8/28/2013; After: 9/11/2013–3/29/2014.

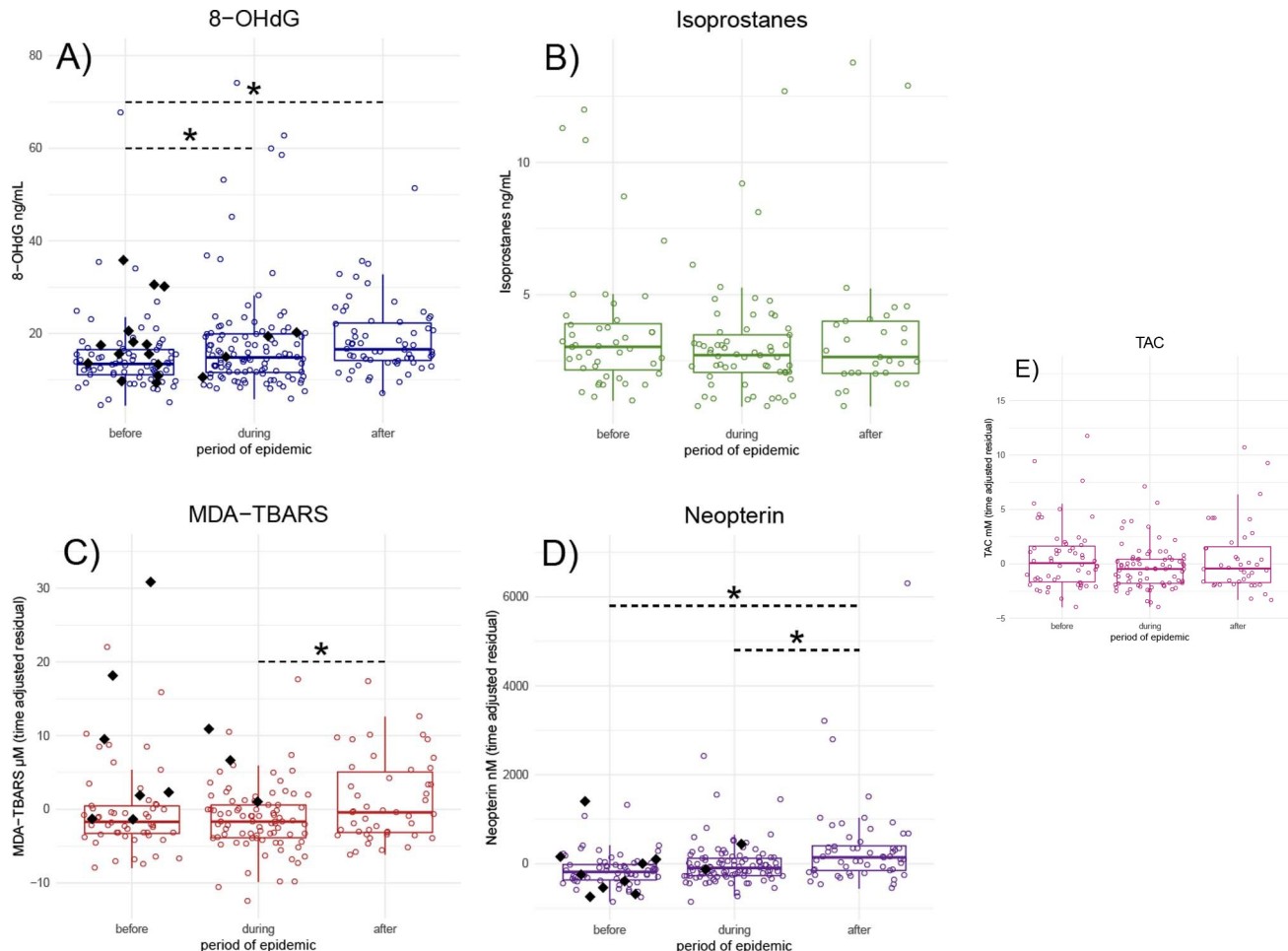

**Fig 2.** Variation in concentrations of A) 8-OHdG, B) F-Isoprostanes, C) MDA-TBARS, D) neopterin, and E) Total antioxidant capacity before, during, and after a severe respiratory epidemic. Points are individual sample-day concentrations. Black diamonds are sample-days from any of the 5 individuals that died during the epidemic. Boxes represent median and interquartile ranges of concentrations. Horizontal dashed lines indicate significant contrasts in biomarker concentrations between periods determined by generalized linear mixed effects models. For N individuals and samples per biomarker see S1 Table.

and MDA-TBARS, as well as 8-OHdG prior to her injury. It is possible that GG was suffering an unknown condition despite having exhibited no other visible clinical signs over the pre-injury period.

## Validation 3. Mixed-longitudinal analysis

Within a mixed-longitudinal sample of individuals spanning all of adulthood, 8-OHdG decreased weakly but significantly with age (Fig 4; GLMM; $n_{individuals, samples}$ = 36, 583; ß ± se = -0.1 ± 0.05, 95% CI = -0.2 –-0.01, p = 0.03). These effects were not significantly different for males and females. No other markers (isoprostanes, MDA-TBARS, neopterin, TAC) varied with age, sex, or age moderated by sex in the mixed-longitudinal sample (S6 Table).

Among the subset of past-prime individuals (35+ yrs), MDA-TBARS increased with age (GLMM; $n_{individuals, samples}$ = 9, 74; ß ± se = 0.09 ± 0.04, 95% CI = 0.2–0.16, p = 0.01; Table in S7 Table and S1B Fig). Further, isoprostanes and TAC changed with age in a sex-specific way, such that both increased with age in past-prime females and decreased among males

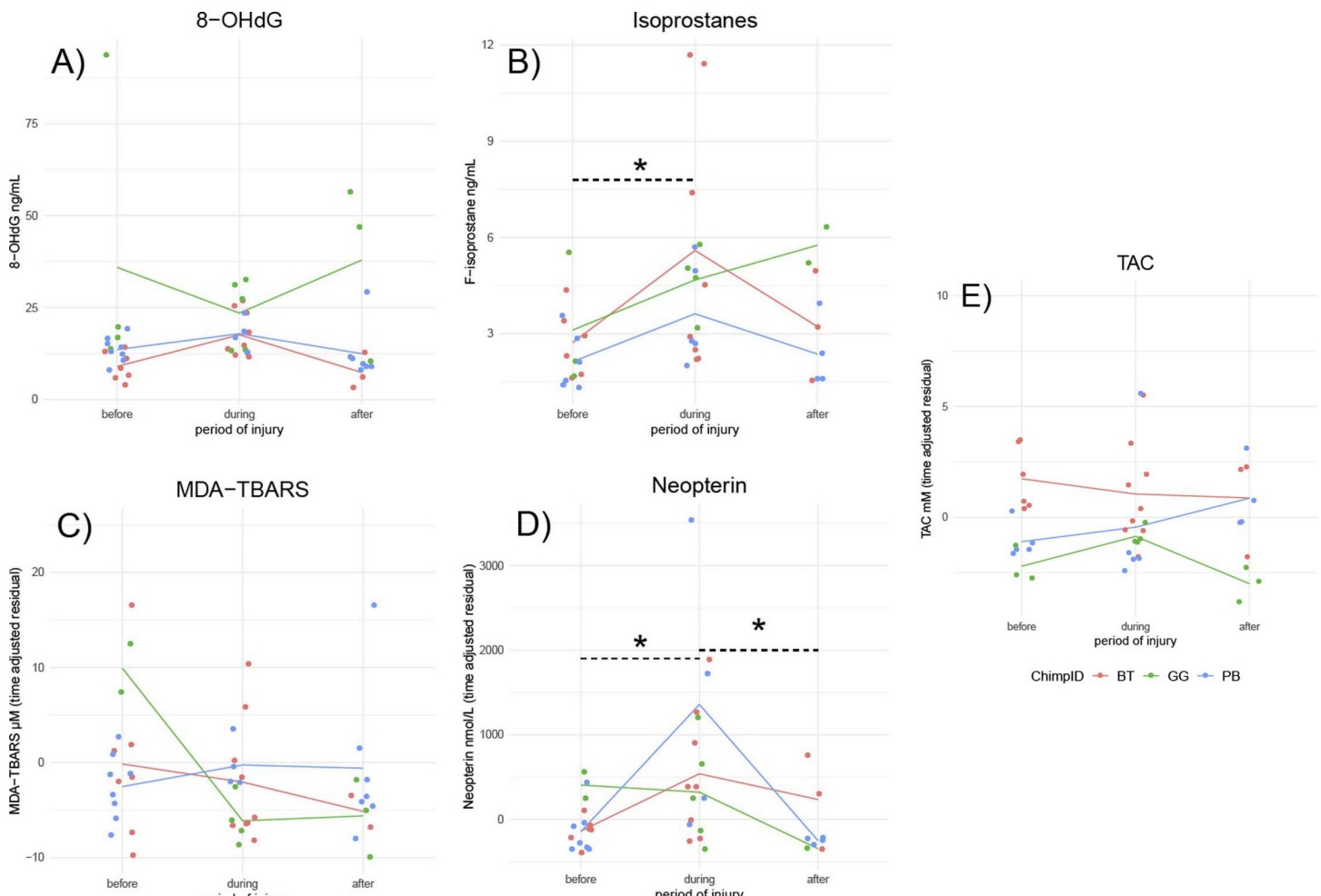

**Fig 3. Variation in OS biomarkers according to stage of severe injury.** B) Isoprostanes and D) neopterin varied significantly according to periods of before, during, and after severe injury. *Asterisks indicate significant contrasts between periods, as determined by glmer models. For N individuals and samples see S2 Table.

(Isoprostanes, GLMM n $_{individuals, samples}$ = 6, 71; ß ± se = -0.55 ± 0.15, 95% CI = -0.84 –-0.26, p < 0.001: TAC, n $_{individuals, samples}$ = 6, 77; ß ± se = -1.96 ± 0.64, 95% CI = -3.22 –-0.07, p = 0.003; S7 Table and S1A and S1C Fig). Neither 8-OHdG and neopterin varied with age, sex, or their interaction in the past-prime mixed-longitudinal sample (S7 Table).

## Validation 4. Lead up to death

For individuals who died during the study period, isoprostanes increased as time to death decreased (Figs 5B and S1A, glmer ß = 0.1, 95% CI 0.01–0.2, p = 0.03). 8-OHdG decreased as individuals approached death (Fig 5A and S8A Table, glmer ß = -0.11, 95% CI -0.18 - -0.05, p = 0.001). Neither neopterin, MDA-TBARS, nor TAC exhibited a significant change over time (Fig 5C–5E). However, inspection of the data suggested that the individual who died in its prime (KK, 27.9 years at death) sometimes exhibited a different pattern than the other 3, who died at an advanced age. Indeed, in models that included only past-prime individuals (OU, BL, ST), neopterin increased as individuals approached death (Fig 5C and S8B Table,

**Table 3. Variation in OS biomarkers between periods of before, during, and after severe injury.** Betas and standard deviations of contrasts from generalized linear mixed effects model and percentage of overall variance explained by individual ID as a random effect. Betas of significant contrasts in bold. n = 3 individuals (2 M, 1 F).

| OS Biomarker | n samples | Before → During | Before → After | During → After | Sex (M) | Age | % RE |
|---|---|---|---|---|---|---|---|
| 8-OHdG | 50 | *0.29 ± 0.17* † | -0.05 ± 0.18 | *-0.34 ± 0.19* † | **-0.85 ± 0.17** *** | 0.23 ± 0.19 | 0 |
| Isoprostanes | 41 | **0.59 ± 0.16** *** | 0.28 ± 0.19 | -0.31 ± 0.19 | *-0.34 ± 0.18* † | **-0.38 ± 0.18** * | 0 |
| MDA-TBARS | 46 | -0.23 ± 0.18 | -0.25 ± 0.21 | -0.02 ± 0.21 | 0.11 ± 0.2 | 0.05 ± 0.21 | 0 |
| Neopterin | 41 | **0.8 ± 0.18** *** | -0.03 ± 0.23 | **-0.82 ± 0.23** *** | 0.26 ± 0.21 | 0.34 ± 0.2 | 0 |
| TAC | 42 | 0.11 ± 0.13 | 0.04 ± 0.15 | -0.07 ± 0.14 | **0.53 ± 0.13** *** | **-0.31 ± 0.14** * | 0 |

† p < 0.10,

* p < 0.05, ** p < 0.01,

*** p < 0.001. Sampling date ranges per individual in S2 Table.

lmer ß = 113.6, 95% CI 28.8–198.4, p = 0.01). The exclusion of the young subject did not affect our results for the other markers (S8A and S8B Table).

## Discussion

In this study, we examined a set of urinary biomarkers to determine their utility for assessing oxidative stress in future studies of health and aging in wild chimpanzees (Table 4). Redox dynamics in the body are highly complex and, accordingly, our results were not always straightforward. We found that DNA damage (urinary 8-OHdG) correlated with all other measures of oxidative damage (isoprostanes, MDA-TBARS) and inflammatory activity (neopterin), and each damage marker responded significantly to one or both natural experiments in which chimpanzees faced significant acute immunological challenges. This suggests overall that urinary markers can provide a useful signal of intracellular oxidative damage induced during immunological challenges in a natural environment. Our present study was powered to detect medium to large effect sizes in its various analyses; we encourage investigators to carefully consult Brysbaert & Stevens [95] in designing future studies with repeated sampling per subject. For future studies that aim to detect patterns of aging, we recommend balancing breadth vs. density when sampling across vs. within adults and, when possible, to maintain sampling over the long-term. Studies of the effects of stressors might aim for dense within-individual sampling within a short period of time.

Our results further indicate that different markers may be affected by different types of stressors and along different time courses. Specifically, although both respiratory and open-wound infections produced inflammation, the respiratory virus primarily led to prolonged DNA damage, while open wounds led to acute increases in lipid peroxidation (isoprostanes). Though the precise mechanisms are unclear, different pathogens lead to different types of damage depending on the immune pathways activated and the cellular sites affected [96,97]. Such findings lead us to recommend that using multiple biomarkers is the most prudent approach to evaluate oxidative damage, particularly if research involves health monitoring that is not specific to one particular type of stressor.

Finally, while theory posits that oxidative stress should increase rather steadily with age, we did not find this. Age-related changes in oxidative markers were only observable in individuals that were either past-prime or in their final years before death, and even then were observed inconsistently. In the mixed-longitudinal sample, individuals > 35 years old showed increased lipid peroxidation with age (MDA-TBARS, and isoprostanes only among females) and changes in TAC where only males followed the expected decline. Among chimpanzees that died, lipid

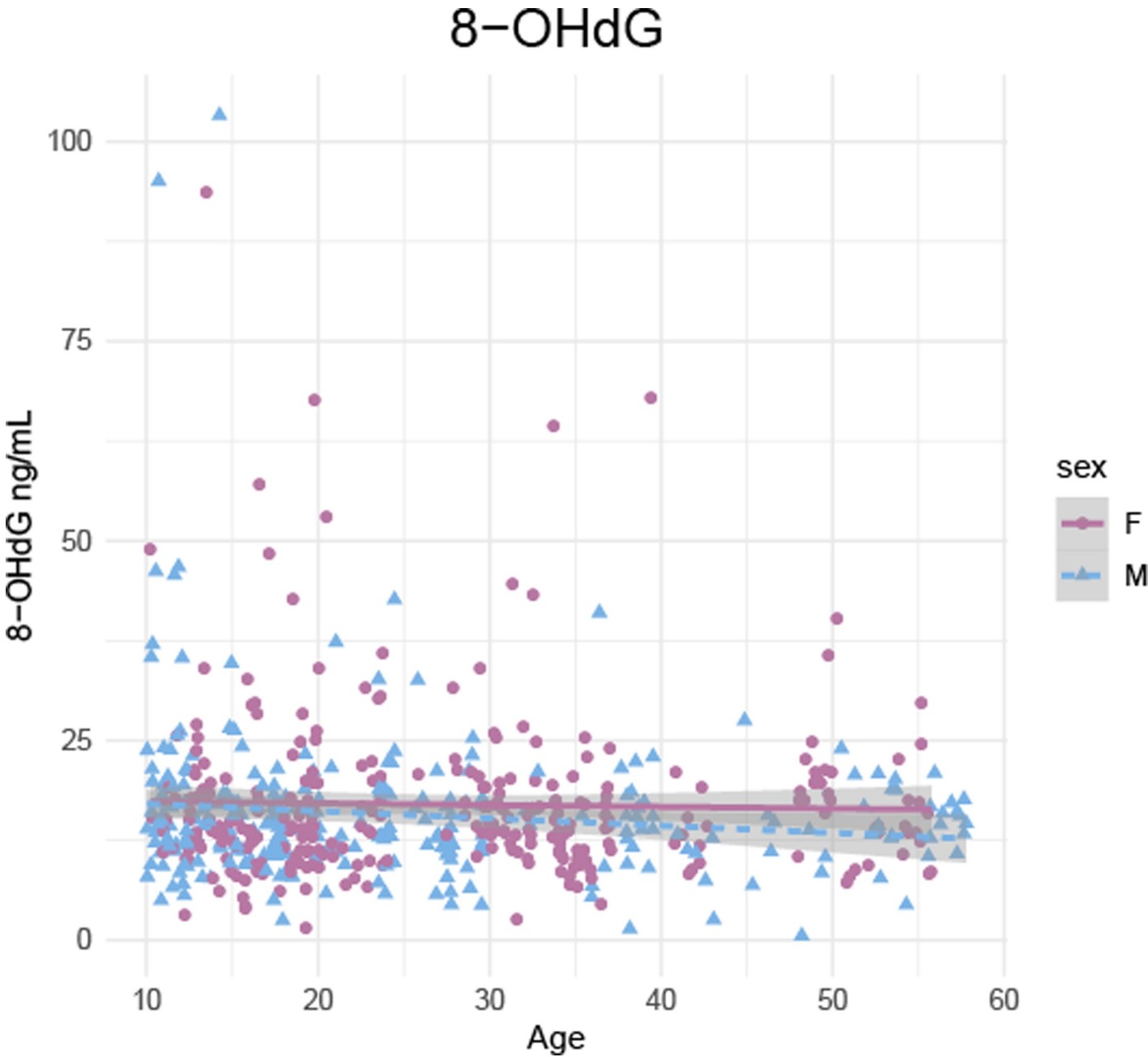

**Fig 4. Decrease in 8-OHdG with age in a mixed-longitudinal sample (n = 36 individuals, 583 samples).**

peroxidation (isoprostanes) and inflammatory activity (neopterin) were observed to increase in the years preceding death. Lastly, one marker, DNA damage (urinary 8-OHdG) actually declined with age in both mixed-longitudinal and within-individual analyses. These findings suggest that while oxidative stress may play a role in declining health during senescence, wild chimpanzees do not experience increases in oxidative damage throughout adulthood. Such a lack of global increases in OS with age aligns with limited findings in other wild animals [10,22,71]. A similarly complex picture is emerging from studies in humans, where age-related increases in oxidative damage and decline in protection often appear throughout adulthood in

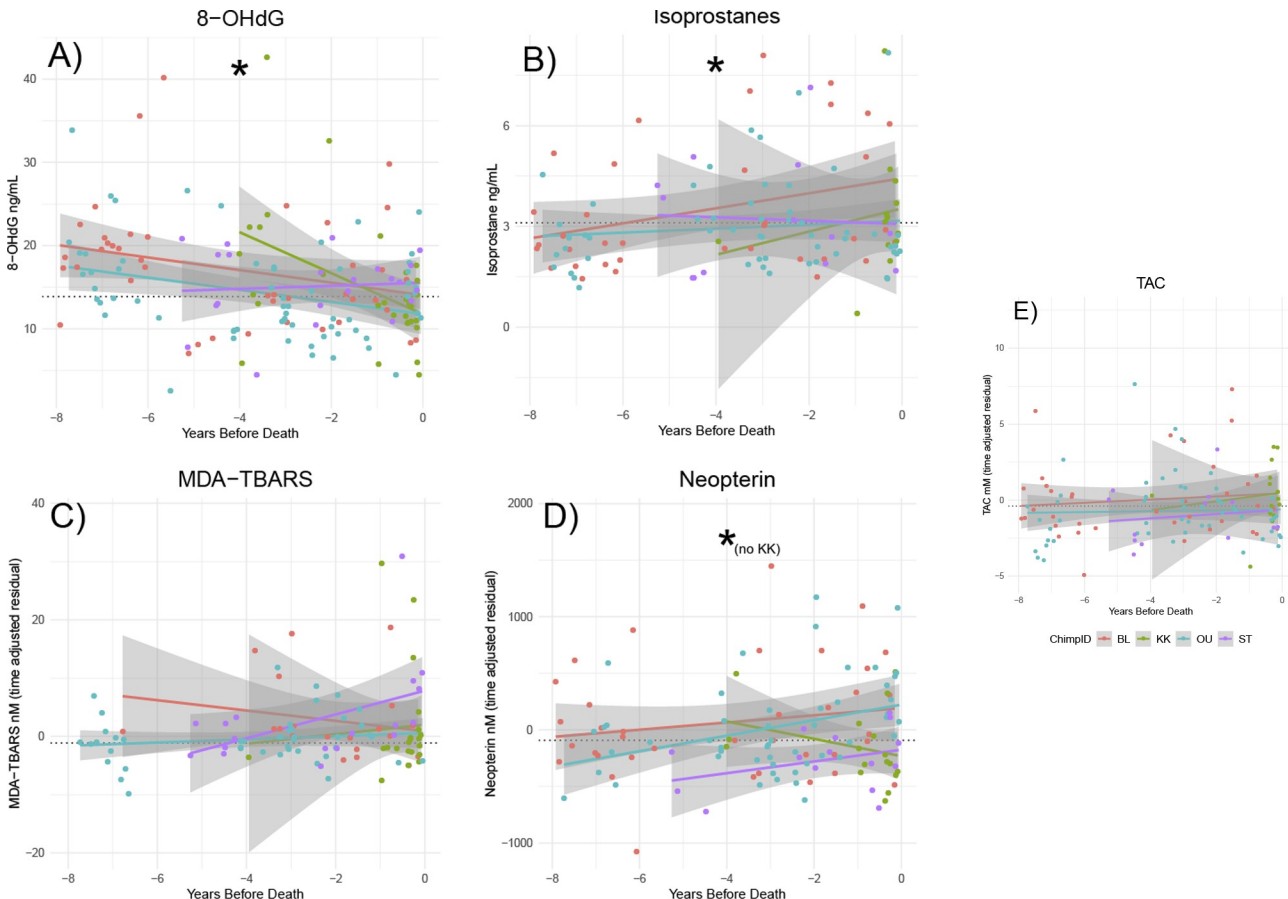

**Fig 5. Variation in OS biomarkers by year before individuals' death.** *A) 8-OHdG decreased and B) Isoprostanes increased among the 4 subjects as they approach death. D) Neopterin increased among past-prime individuals as they approached death. Solid line drawn with simple linear regression, unlike mixed effect models that determined significance. Black dotted line represents population median in biomarker from samples unrelated to epidemic, injury or eventual death.

cross-sectional and mixed-longitudinal studies [69,70,86,98–102], but not always [23,55,103–106]. For example, urinary 8-OHdG increased with age in all patients with muscular dystrophy [86] and between certain age categories in a large cross-sectional sample of men and women [100], but did not increase with age in pollution-exposed silica miners [106].

**Table 4. Summary of findings.**

| Marker | Respiratory infection | Open wound injury | Age | Years before death |
|---|---|---|---|---|
| 8-OHdG | ↑ During & After | - | ↓ | ↓ |
| Isop | - | ↑ During | ↑$_F$ ↓$_M$[†] | ↑ |
| MDA-TBARS | ↑ After | - | ↑[†] | - |
| Neopterin | ↑ After | ↑ During | - | ↑* |
| TAC | - | - | ↑$_F$ ↓$_M$[†] | - |

* In model excluding KK,

[†] In model of past-prime individuals

## General correlations among OS biomarkers

Damage to DNA (8-OHdG) positively correlated with lipid peroxidation (isoprostanes, MDA-TBARS) and macrophage activity (neopterin). This result was expected because reactive oxygen species, including hypochlorous acid, peroxynitrite, and the hydroxyl radical, cause damage to both DNA and lipids and are produced in large quantities by activated phagocytic cells, such as macrophages [4,6,28]. Surprisingly, however, neither isoprostanes nor MDA-T-BARS correlated with neopterin. Generally, the modest associations across markers suggest that they each provide independent information on oxidative status. As our experiments suggest, this may be due to variable pathways of oxidative damage [96] or to differences in time scales of response.

We predicted that total antioxidant capacity (TAC) would be negatively associated with markers of damage because it is this imbalance that characterizes oxidative "stress". As damage-causing radicals increase, more antioxidants should be depleted, as during activities such as exercise ([107]). In fact, however, we found that TAC correlated positively with lipid damage (isoprostanes) and inflammation (neopterin). One reason that TAC may increase alongside markers of damage is that pro-oxidants activate Nrf2 transcription factor, which increases the expression of endogenous antioxidant enzymes [108,109]. Nrf2 has evolved such that common processes that produce reactive oxygen species, e.g. physical activity and critical illness, also adaptively induce production of antioxidants such as glutathione and uric acid [110–113]. Overall, we found little evidence to suggest that antioxidant capacity was limiting in chimpanzees, either during acute stressors or during aging.

## Acute responses to infection

Infection from both respiratory disease and severe injury led to changes in neopterin and either in markers of DNA damage (8-OHdG) or in lipid peroxidation (isoprostanes). Reactive oxygen species are generated in large numbers during the oxidative burst of macrophages [28,30]. Accordingly, changes in neopterin levels paralleled those of the main OS marker that responded to each type of infection, respiratory (8-OHdG) and open wounds (isoprostanes).

In other non-human primates assayed during infections, neopterin concentrations are clearly elevated relative to baseline, both in captive [46,45] and wild animals [47], and often peak during a short window [83]. For example, among chimpanzees in the Taï National Park, Cote D'Ivoire, neopterin was elevated during an acute viral respiratory outbreak and declined to baseline levels after signs of respiratory infection diminished [47]. Similarly, in rhesus macaques experimentally inoculated with SIV, concentrations in neopterin peaked sharply over 2–3 days approximately 2 weeks following infection, after which concentrations declined but remained elevated from baseline for several weeks [83]. The fact that neopterin appears to spike briefly, even when the infectious agent persists, means that it can be difficult to capture this signal. At Kanyawara, we did not observe a significant increase in neopterin during the course of a viral respiratory outbreak, despite relatively high sampling intensity. However, we did observe increased levels in the period after the respiratory outbreak. By contrast, neopterin levels increased acutely when chimpanzees had severe open wounds and returned to levels that were close to baseline after injuries were healed. This acute response in neopterin strongly suggests the presence of infection, as neopterin was not found to spike in response to surgical trauma alone in captive rhesus macaques [114].

Unlike the short-term and acute respiratory infections monitored in Taï chimpanzees and rhesus macaques, the respiratory epidemic in the Kanyawara chimpanzee community lasted for 4 months, with three waves of respiratory signs suggesting possible reinfection [90]. Indeed, 47 of 53 community members were seen coughing and 31 had productive coughs

during multiple months of 2013, and deaths occurred over a span of three months. While it is surprising that our opportunistic sampling procedure failed to capture the expected spikes in neopterin during the outbreak, the elevation in levels after the outbreak suggest that such prolonged (or repeated) illness exerts somatic costs that far outlive the active infection. Whereas abrupt increases in neopterin can be useful to identify infections (in cases that might not be as obvious as respiratory illnesses), it is these lasting inflammatory effects, related to increased oxidative damage, that are predicted to be of particular relevance to longevity. Accordingly, 8-OHdG was increased in our chimpanzees not only during but after the respiratory outbreak.

## Variation in damage with age and senescence

Our analysis of oxidative status with age yielded complex results, such that different markers produced effects during different life stages and at different time scales. In a mixed-longitudinal sample spanning all of adulthood, we found no evidence for increases in oxidative damage and inflammatory activity, nor a decrease in antioxidant capacity with chronological age. However, in past-prime individuals alone, MDA-TBARS increased with age, while isoprostanes and TAC increased among females and decreased in males. Female chimpanzees' unexpected rise in TAC with age may have resulted from reproductive events that trigger enhanced antioxidant protection, as seen in some species [26,27].

In contrast to both mixed-longitudinal analyses, neopterin did increase among very old chimpanzees in their years just prior to their deaths. One possible reason for different signals among our mixed-longitudinal and within-individual age-based analyses is selection bias, as chimpanzee populations have higher variation in mortality than human populations typically do [81,79,115]. Only 18% of chimpanzees at Kanyawara live until age 30, whereas 42% of Hadza hunter gatherers reach age 30 [115]. Thus, even if oxidative stress increases, individuals present in the dataset at later ages could exhibit relatively low oxidative burden compared to those that died at younger ages. Additionally, our within-individual analysis featured denser sampling which may have increased our ability to detect small effects. It appears that while chimpanzees experience acute periods of oxidative stress across the life course they, like many humans, may not suffer a chronic and global redox imbalance unless they reach old age and/or are experiencing critical declines in health. In either case, we lack evidence that cumulative oxidative damage is a typical component of chimpanzee aging, despite long lifespans.

The inflammation signal of immunosenescence in chimpanzees appears weaker than it does in humans, where a rise in inflammation with age is prominent in both longitudinal and cross-sectional studies. Indeed, in humans increased inflammatory activity is a core component of aging that is visible cross-sectionally in both post-industrial [31,32,48,98,116] and some indigenous horticultural populations [36]. The within-individual increase in lipid peroxidation and inflammation in past-prime chimpanzees preceding death could, however, suggest a late-emerging pattern of inflammaging. Namely, on top of an already immunosenescent profile, cellular debris from oxidative damage could perpetuate an inflammatory immune response [31,98].

An unexpected finding was a decrease in urinary 8-OHdG in association with senescence. This pattern was weakly expressed in the mixed-longitudinal data but emerged quite strongly in the years shortly before death. There are several plausible mechanisms to explain such a pattern. First, declining 8-OHdG could result from declining energy metabolism with age, e.g. decreased reproductive effort and reduced physical activity [22,55,117]. For example, urinary 8-OHdG decreased with age in a sample of male rhesus macaques, which mirrored a decline in reproductive effort [22]. However, contradictory evidence from human and rodent studies show that correlations of oxidative damage with resting metabolic rate and reproductive

investment are absent or negative, respectively, possibly because damage induces the expression of protective antioxidant enzymes [10,55]. A second reason for the age-related decline in 8-OHdG is perhaps because its excretion into urine is actually dependent on nucleotide excision, a key process involved in repairing DNA lesions [51,52,106,118]. While assays of urinary 8-OHdG are commonly used, often under the assumption that repair processes are maximally efficient, nucleotide excision repair in fact declines with age in both humans and mice [15,17]. Relatedly, studies in rats have found that while 8-OHdG in organs and blood cells increased with age, urinary 8-OHdG decreased [14].

## Limitations

We acknowledge certain limitations of this study. Our natural experiments were simply that and accordingly limited in scope. Due to repeated sampling, such conditions were sufficient for statistical evaluation, but sampling of acute injuries was limited to few individuals and a larger sample assayed over time is necessary to characterize normative changes in redox status with age. Further, many other potential causes of variation in oxidative stress, such as the effects of social and reproductive factors, as well as specific causes of death, were unexplored in this study. These are worthy of future examination but require more specifically tailored sample selection, e.g. dense sampling before, after, and during pregnancy and lactation. Because chimpanzees do not exhibit reproductive seasonality, and because we analyzed repeated samples from individuals spread over long periods of time, we have no indication that our sample is biased in relation to reproduction or any other sociodemographic factor. We also acknowledge that non-invasive measures are often limited in their specificity. Currently, for example, DNA damage and repair cannot be measured directly and independently of one another in urine (e.g., [119,120]). Also, general antioxidant capacity measures allow particularly abundant antioxidants, such as uric acid, to drown out nuanced signals in other antioxidants [73,103,109,121]. Non-invasive markers should thus be interpreted as indicators rather than direct measures of physiological processes.

## Conclusions

Urinary markers of DNA damage and lipid peroxidation, specifically, 8-hydroxy-2'-deoxyguanosine and 15-Isoprostane $F_{2t}$, are particularly useful, alongside inflammatory markers (neopterin), to monitor acute oxidative stress and its potential role in age-related declines in health and mortality in wild primates. We advise measuring both DNA and lipid markers simultaneously, as they may indicate different pathways of oxidative damage resulting from different pathogenic challenges and age.

Although a larger sample is required to authoritatively delineate age-related patterns of OS and immunosenescence in chimpanzees, we found no evidence that oxidative stress generally accumulates or that immune regulation generally declines with age in this species. Oxidative damage and inflammation do, however, appear to be associated with healthspan, as they increase in particularly old individuals and those that are approaching death. Such findings broadly warrant further research on apes and non-WEIRD humans (i.e. Western, Educated, Industrialized, Rich, and Democratic; Henrich et al., [122]) to identify if the common occurrence of inflammaging in humans is evolutionarily aberrant. Lastly, the limitation of urinary 8-OHdG as a marker of both DNA damage and repair emphasizes the value of both current invasive sampling to characterize age-related changes in DNA damage, and future development of assays for DNA repair processes in non-invasive samples.

## Supporting information

**S1 Table. Sampling per biomarker before, during, and after respiratory epidemic.**
(DOCX)

**S2 Table. Individual sampling per biomarker before, during, and after severe injury.**
(DOCX)

**S3 Table Individual sampling ranges A) per year and B) per biomarker per year leading up to death.**
(DOCX)

**S4 Table. Effect of time of day on biomarkers.** Models include only samples from individuals not during or after epidemic or injury. Significant relationships in bold.
(DOCX)

**S5 Table Correlations between OS biomarkers, assessed with generalized linear mixed effects models with individual ID as a random effect Significant relationships in bold.**
(DOCX)

**S6 Table. Cross-sectional variation in OS biomarker by individual age and sex.** Betas and standard deviations of predictors from generalized linear mixed effects model and percentage of overall variance explained by individual ID as a random effect. Age by sex interactions in grey shading, extracted from separate model that included main effects of age and sex. Significant effects in bold.
(DOCX)

**S7 Table. Cross-sectional variation in OS biomarker of past prime individuals ($>$ 35 years old) by age and sex.** Betas and standard deviations of predictors from generalized linear mixed effects model and percentage of overall variance explained by individual ID as a random effect. Age by sex interactions in grey shading, extracted from separate model that included main effects of age and sex. Significant effects in bold.
(DOCX)

**S8 Table. Variation in OS biomarkers among subjects that died during observation over years leading up to death.** Models A) including and B) excluding relatively young individual KK. Betas and standard deviations of predictors from linear (neopterin and TAC) and generalized linear mixed effects model and percentage of overall variance explained by individual ID as a random effect. Significant effects in bold.
(DOCX)

**S1 Fig. Changes in biomarkers with age among past-prime individuals ($>$ 35 years old) in a mixed-longitudinal sample.** A) Isoprostanes increase with age among females and decrease among males (n = 6 individuals, 71 samples); B) MDA-TBARS increases with age (n = 9 individuals, 74 samples); C) TAC increases with age among females and decreases among males (n = 6 individuals, 77 samples).
(TIFF)

## Acknowledgments

We thank the staff and field assistants of the Kibale Chimpanzee Project for collecting samples and daily records of health status. The authors also thank the Uganda Wildlife Association, the Uganda National Council for Science and Technology, and Makerere University Biological

Field Station for their support in conducting long term research in Kibale National Park. Kris Sabbi, Seth Merkley, and Megan Cole provided assistance in the laboratory.

## Author Contributions

**Conceptualization:** Melissa Emery Thompson.

**Data curation:** Emily Otali, Zarin Machanda, Martin N. Muller, Richard Wrangham.

**Formal analysis:** Nicole Thompson González.

**Funding acquisition:** Melissa Emery Thompson.

**Investigation:** Nicole Thompson González, Emily Otali, Zarin Machanda, Martin N. Muller, Richard Wrangham, Melissa Emery Thompson.

**Methodology:** Nicole Thompson González, Richard Wrangham, Melissa Emery Thompson.

**Project administration:** Emily Otali, Zarin Machanda, Martin N. Muller, Richard Wrangham.

**Resources:** Richard Wrangham, Melissa Emery Thompson.

**Supervision:** Melissa Emery Thompson.

**Validation:** Nicole Thompson González.

**Visualization:** Nicole Thompson González.

**Writing – original draft:** Nicole Thompson González.

**Writing – review & editing:** Nicole Thompson González, Zarin Machanda, Martin N. Muller, Richard Wrangham, Melissa Emery Thompson.

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
