## [Decision Letter · Decision Letter 0]

8 Jun 2020

PONE-D-20-04637

Urinary markers of oxidative stress respond to infection and late-life in wild chimpanzees

PLOS ONE

Dear Dr. Thompson,

Thank you for submitting your manuscript to PLOS ONE. After careful consideration, we feel that it has merit but does not fully meet PLOS ONE’s publication criteria as it currently stands. Therefore, we invite you to submit a revised version of the manuscript that addresses the points raised during the review process.I strongly recommend to assess the proportion of variability explained by your fixed and random terms included in your GLMM (e.g., using the *MuMIn *package) to better understand the individual contribution of your observations in your set or oxidation biomarkers. On the other hand, I do not understand why you did not include the age factor (e.g., in months as covariate) in your infection models. The #3 referee has also raised this age-related issue. In the attached file, you will get more comments on your work as well as other very valuable recommendations made by our reviewers.

We look forward to receiving your revised manuscript.

Kind regards,

Emmanuel Serrano, PhD

Academic Editor

PLOS ONE

Journal Requirements:

Reviewers' comments:

Reviewer's Responses to Questions

**Comments to the Author**

1. Is the manuscript technically sound, and do the data support the conclusions?

Reviewer #1: Yes

Reviewer #2: Yes

Reviewer #3: Yes

2. Has the statistical analysis been performed appropriately and rigorously? 

Reviewer #1: I Don't Know

Reviewer #2: Yes

Reviewer #3: Yes

3. Have the authors made all data underlying the findings in their manuscript fully available?

Reviewer #1: Yes

Reviewer #2: Yes

Reviewer #3: Yes

4. Is the manuscript presented in an intelligible fashion and written in standard English?

Reviewer #1: Yes

Reviewer #2: Yes

Reviewer #3: Yes

5. Review Comments to the Author

Reviewer #1: Dear Dr. Serrano,

The present study examines the use of urinary oxidative stress (OS) markers for monitoring both acute, short term health challenges and long-term patterns of aging in free-living chimpanzees. Monitoring the health status of free-living non-human primates may be challenging due to multiple factors such as inhabiting remote areas, brief clinical expression and survival behaviours, and especially because direct sampling from moribund or alive animals may be not feasible. Thus, new non-invasive approaches are needed, as well as the development and validation of tests for these species in the wild. An integration of expertise from different professional sectors of the health sciences is essential to address questions to wildlife health, and in particular, cooperation with human doctors and incorporation of human medicine techniques for assessing health status in chimpanzees is invaluable. Given all of these considerations, the present study is of great interest.

I consider this article is suitable for publication, although there are some weaknesses that should be addressed:

MINOR ISSUES

1. Objectives: the aim/objectives of the study should be easily identified before “Methods” section, together with hypothesis/expectations. It should not be duplicated as it happens in lines (51-53) in the ‘Introduction’ section and later on the ‘Study system’ section (lines 165-171). This could lead to confusion and gives a disorganized overview.

2. Methods:

- ‘Before, during and after serious injury’ section: it should be clarified whether these 3 chimpanzees were not sampled during the respiratory illness epidemic, as it is stated in the ‘Quarterly mixed-longitudinal’ study section.

3. Limitations: although statistically representative outcomes, sample size may not be representative for the assessment of acute challenge response such as severe traumatic injury (n=3) and neither for ‘lead up to death’ variation (n=4). Also, it may be interesting to point out that ‘unknown’ causes of death from the ‘lead up to death’ study may lead to response bias.

FORMAT

1. Line 414: add spacing ‘One individual’.

2. Review the ‘References’ section format: duplicity, capital letters (lines 989-990), scientific names in non-italic format (line 721-722), etc.

Reviewer #2: This is what it is. A useful addition to the literature on biomarkers of oxidative stress in primates under physiologically relevant conditions. The study of wild animals is laudable. The limitations of the study are appropriately acknowledged.

Reviewer #3: General comments:

This study explores 5 different potential non-invasive markers of oxidative stress in chimpanzees. This study utilizes an impressive sample size, with densely sampled longitudinal data than is rare for non-human primate studies, particularly those of wild populations. We know that oxidative stress is an important part of the aging process in humans and occurs during times of immune challenge, suggesting that non-invasive markers of oxidative stress could be a useful tool for ways monitoring the health status and progressive aging of wild primate populations.

This study offers a biological validation for the OS markers in question, although the markers do not always respond in expected ways. This reinforces the need for analytical validation of these assays in chimpanzees, which is essential to understanding the specificity and sensitivity of the assay. I am sure this is in part due to the ethics of invasive research with chimpanzees, and so alternative approaches to determine the sensitivity and cross reactivity of the assay (such as serial dilutions or spiking of pooled samples with standard) would help to reassure that the assays are indeed measuring the desired compound and only the desired compound.

Notably, this study examines urine samples from a densely sampled and closely monitored population, and therefore is able to distinguish small differences in marker excretion between experimental subgroups. As most studies of wild primates are not able to consistently obtain samples from such a large number individuals over a long time frame, a discussion of necessary sample sizes would be useful in determining how feasible it would be to identify meaningful changes in these biomarkers in other less accessible populations.

In the abstract, the author states that 5 urinary markers of oxidative damage were assessed, however the author only names four of the five markers. TAC should be mentioned in some capacity in the abstract as well to avoid this confusion.

The author also mentions that non-invasive markers of OS may be useful in exploring the costs associated with life history investments, which leads me to wonder how these markers of OS respond to pregnancy and lactation, and if any reproductive parameters were assessed in relationship to OS markers in this study sample. Including sex as a random effect in the models could account of some variance due to female reproductive state, but could demographic changes in the number of pregnant or lactating females confound time period comparisons? A discussion of female reproductive state in your sample would be appreciated.

I also believe that the grouping of samples by time-period instead of by individual health status is potentially problematic. Is more refined data on which individuals were sick and when available? The health of this study population seems to be closely recorded, and if this data is available, the authors should compare sick individuals with healthy individuals, instead of grouping samples by time period without knowing individual health status. The before and after periods are quite long, and it seems reasonable that individuals that experienced illness during the epidemic phase could have also experienced a separate illness during the before or after phase, potentially muddying the comparisons.

In humans, does OS stress increase linearly with age from the time of birth? Or does it only start to increase after a certain old age? I wonder if by looking at such a wide range of ages, there may be some signals of aging that are lost if they only occur later in life. I understand this is kind of what the “Years before death” analysis is getting at, but it may be interesting to conduct a similar age and OS analysis including only samples from individuals above a certain older age.

Overall, this study offers an excellent discussion of potential urinary biomarkers of OS in non-human primates, and gives preliminary support for their utility in monitoring wild primate populations. The results regarding health status and OS markers could be made clearer by utilizing individual health status as opposed to group trends, if that data is available. Additionally, some of the language used in the MS is not well defined, and could be more easily understood with clearer definitions throughout. In sum, this study is a valuable addition to the literature on OS in primates.

Line-by-line comments:

Line 18- It seems as though the results of this study do not support OS as playing a central role in aging in chimpanzees, so this is an interesting choice of first sentence.

Line 38- Please briefly explain what is meant by “oxygen’s reactive species”

Line 43- “thus” implies that the second half of the sentence is a logical conclusion of what precedes it, and I don’t think this is the case for this statement.

Line 46- MDA-TBARS needs to be defined before using its acronym.

Line 89- Please explain what is meant by ‘redox status’. It is unclear if this is the same thing as OS.

Line 114- This explanation makes it sound like 8-OHdG is actually a signal of DNA repair and not DNA degradation. Are these two processes always directly proportional?

Line 120- The author previously mentions the oxidation of lipid as an OS marker, but now refers to it as lipid peroxidation. Are these the same thing?

Line 124- Are these other physiological processes known? It would be useful to know what other processes are associated with MDA excretion in order to fully understand the results.

Line 141- What is meant by controlled substances?

Line 144- What is meant by inducability of expression? I am not familiar with that terminology.

Line 175- You define MDA earlier, but “TBARS” has still not been defined.

Line 235- This full definition of MDA-TBARS comes very late in the MS.

Line 241- After visually inspecting the data “for” inflated biomarker values.

Line 342- Why not include time of day as a covariate instead of using residuals as data? Although it is common practice, using residuals as data is thought to lead to biased parameter estimates. If you must, please provide strong reasoning for doing so. Was there considerable agreement across the population for the population slope? See:

Freckleton, R. P. The seven deadly sins of comparative analysis. J. Evol. Biol. 22, 1367–1375 (2009).

Line 356- What is the response variable in this model? The composition of each model needs to communicated in a much clearer way, such as numbering them or writing them out in a table.

Line 411- This is interesting because a recent study did not observe rises in uNEO in response to surgical trauma. Please see and discuss this discrepancy:

Higham, J. P., Stahl-Hennig, C., & Heistermann, M. (2020). Urinary suPAR: a non-invasive biomarker of infection and tissue inflammation for use in studies of large free-ranging mammals. Royal Society open science, 7(2), 191825.

Line 414- “One individual”

Line 607- What does RMR stand for?

Line 644- Is WEIRD an acronym here or are you just saying that we are weird?

Line 1375- Reference all capitalized

6. PLOS authors have the option to publish the peer review history of their article (what does this mean?). If published, this will include your full peer review and any attached files.

Reviewer #1: No

Reviewer #2: No

Reviewer #3: Yes: Rachel Petersen

---

## [Author Response · Author response to Decision Letter 0]

23 Jul 2020

PONE-D-20-04637

Urinary markers of oxidative stress respond to infection and late-life in wild chimpanzees

PLOS ONE

Dear Dr. Thompson,

Thank you for submitting your manuscript to PLOS ONE. After careful consideration, we feel that it has merit but does not fully meet PLOS ONE’s publication criteria as it currently stands. Therefore, we invite you to submit a revised version of the manuscript that addresses the points raised during the review process.

I strongly recommend to assess the proportion of variability explained by your fixed and random terms included in your GLMM (e.g., using the MuMIn package) to better understand the individual contribution of your observations in your set or oxidation biomarkers. 

AUTHORS: We have added a column to all GLMM results tables to indicate the proportion of variance explained by individual ID as a random effect. Changes were made to Tables 2 & 3, and Supplementary Tables 6 – 8.

On the other hand, I do not understand why you did not include the age factor (e.g., in months as covariate) in your infection models. The #3 referee has also raised this age-related issue. 

AUTHORS: We included age in our original infection models and now clarify the structure of each of our statistical models in revised Table 1.

In the attached file, you will get more comments on your work as well as other very valuable recommendations made by our reviewers.

AUTHORS: We have corrected the relevant issues that the reviewer marked in the submission’s pdf. These include replacing “N” with “n” where relevant and adding the versions of R packages and a citation for the R programming language. We also clarified the comparisons between before, during, and after periods in results Tables 2 & 3, by adding arrows to indicate that the change in marker concentrations was during a shift from one period to the other. 

We appreciated the reviewer’s important comment that oxidative damage is known to decrease with age in certain species and that environmental variation can produce a stronger signal of oxidative damage than infections alone. We address these nuances in two ways. First, we cite and discuss evidence that does not support a simple oxidative theory of aging and life history in Introduction lines 68-75. Second, we now address that individuals’ lifetime experience, which would include ecological environmental variables, was outside of the current study’s scope in Limitations lines 689-691.

We look forward to receiving your revised manuscript.

Kind regards,

Emmanuel Serrano, PhD

Academic Editor

PLOS ONE

Journal Requirements:

AUTHORS: Included now in Methods lines 209-212.

AUTHORS: Done.

Reviewers' comments:

Reviewer's Responses to Questions

Comments to the Author

1. Is the manuscript technically sound, and do the data support the conclusions?

Reviewer #1: Yes

Reviewer #2: Yes

Reviewer #3: Yes

2. Has the statistical analysis been performed appropriately and rigorously?

Reviewer #1: I Don't Know

Reviewer #2: Yes

Reviewer #3: Yes

3. Have the authors made all data underlying the findings in their manuscript fully available?

Reviewer #1: Yes

Reviewer #2: Yes

Reviewer #3: Yes

4. Is the manuscript presented in an intelligible fashion and written in standard English?

Reviewer #1: Yes

Reviewer #2: Yes

Reviewer #3: Yes

5. Review Comments to the Author

Reviewer #1: Dear Dr. Serrano,

The present study examines the use of urinary oxidative stress (OS) markers for monitoring both acute, short term health challenges and long-term patterns of aging in free-living chimpanzees. Monitoring the health status of free-living non-human primates may be challenging due to multiple factors such as inhabiting remote areas, brief clinical expression and survival behaviours, and especially because direct sampling from moribund or alive animals may be not feasible. Thus, new non-invasive approaches are needed, as well as the development and validation of tests for these species in the wild. An integration of expertise from different professional sectors of the health sciences is essential to address questions to wildlife health, and in particular, cooperation with human doctors and incorporation of human medicine techniques for assessing health status in chimpanzees is invaluable. Given all of these considerations, the present study is of great interest.

I consider this article is suitable for publication, although there are some weaknesses that should be addressed:

MINOR ISSUES

1. Objectives: the aim/objectives of the study should be easily identified before “Methods” section, together with hypothesis/expectations. It should not be duplicated as it happens in lines (51-53) in the ‘Introduction’ section and later on the ‘Study system’ section (lines 165-171). This could lead to confusion and gives a disorganized overview.

AUTHORS: Our apologies for the confusion. “Study system” is within the Introduction and not Methods. This layout might be clearer once the article is in publication format. We think Introduction lines 53-56 gives an important initial summary of the study’s aims that we then expand on immediately before the hypotheses and predictions in the last paragraph of Introduction.

2. Methods:

- ‘Before, during and after serious injury’ section: it should be clarified whether these 3 chimpanzees were not sampled during the respiratory illness epidemic, as it is stated in the ‘Quarterly mixed-longitudinal’ study section.

AUTHORS: We have clarified that the 3 injured chimpanzees were not sampled during the respiratory epidemic (Lines 319-320).

3. Limitations: although statistically representative outcomes, sample size may not be representative for the assessment of acute challenge response such as severe traumatic injury (n=3) and neither for ‘lead up to death’ variation (n=4). Also, it may be interesting to point out that ‘unknown’ causes of death from the ‘lead up to death’ study may lead to response bias.

AUTHORS: We amended our limitations section to point out that although the conditions of our natural experiments were statistically representative, acute injuries were limited to few individuals. We also acknowledge that causes of death were not specifically addressed in Limitations revised lines 689-691. 

FORMAT

1. Line 414: add spacing ‘One individual’.

AUTHORS: Done.

2. Review the ‘References’ section format: duplicity, capital letters (lines 989-990), scientific names in non-italic format (line 721-722), etc.

AUTHORS: Thank you for pointing this out. We have fixed all issues.

Reviewer #2: This is what it is. A useful addition to the literature on biomarkers of oxidative stress in primates under physiologically relevant conditions. The study of wild animals is laudable. The limitations of the study are appropriately acknowledged.

Reviewer #3: General comments:

This study explores 5 different potential non-invasive markers of oxidative stress in chimpanzees. This study utilizes an impressive sample size, with densely sampled longitudinal data than is rare for non-human primate studies, particularly those of wild populations. We know that oxidative stress is an important part of the aging process in humans and occurs during times of immune challenge, suggesting that non-invasive markers of oxidative stress could be a useful tool for ways monitoring the health status and progressive aging of wild primate populations.

This study offers a biological validation for the OS markers in question, although the markers do not always respond in expected ways. This reinforces the need for analytical validation of these assays in chimpanzees, which is essential to understanding the specificity and sensitivity of the assay. I am sure this is in part due to the ethics of invasive research with chimpanzees, and so alternative approaches to determine the sensitivity and cross reactivity of the assay (such as serial dilutions or spiking of pooled samples with standard) would help to reassure that the assays are indeed measuring the desired compound and only the desired compound.

AUTHORS: We agree that such measures are important for validation of non-invasive samples. We did test the recovery of the 8-OHdG and Isoprostanes assays by spiking samples with standard. These two immunoassays were not previously tested for recovery in chimpanzee urine, but the assay for urinary neopterin was, so we reference this validation (Behringer et al. 2017, Frontiers in Physiology). Sensitivity was not an issue for any of these assays as detected concentrations were generally well above the minimum. To provide further information on biomarker kit sensitivity, we have added the proportion of samples with assayed values above each kit’s minimum detected concentration in Methods revised lines 266-268. We also added that protocols for isoprostanes and MDA-TBARS included administering agents to increase specificity in Methods revised lines 256-258. Assays for these two markers have been highlighted in past literature as having limited specificity, however current assays are now improved.

Notably, this study examines urine samples from a densely sampled and closely monitored population, and therefore is able to distinguish small differences in marker excretion between experimental subgroups. As most studies of wild primates are not able to consistently obtain samples from such a large number individuals over a long time frame, a discussion of necessary sample sizes would be useful in determining how feasible it would be to identify meaningful changes in these biomarkers in other less accessible populations.

AUTHORS: In Discussion, we encourage readers to carefully consult Brysbaert & Stevens (2018, Journal of Cognition) to determine if their analyses with appropriate power. We do not give specific numbers as power depends on the structure of the analysis (e.g. repeated measures per individual, comparison of means by time period, number of time periods). However, we do offer some general advice to future researchers interested in assessing age-related changes and responses to stressors in lines 519-523.

In the abstract, the author states that 5 urinary markers of oxidative damage were assessed, however the author only names four of the five markers. TAC should be mentioned in some capacity in the abstract as well to avoid this confusion.

AUTHORS: We now mention total antioxidant capacity in the abstract to avoid confusion.

The author also mentions that non-invasive markers of OS may be useful in exploring the costs associated with life history investments, which leads me to wonder how these markers of OS respond to pregnancy and lactation, and if any reproductive parameters were assessed in relationship to OS markers in this study sample. Including sex as a random effect in the models could account of some variance due to female reproductive state, but could demographic changes in the number of pregnant or lactating females confound time period comparisons? A discussion of female reproductive state in your sample would be appreciated.

AUTHORS: We agree with the reviewer that this is an interesting issue. However, we feel this (and other questions about the demographic/behavioral determinants of OS) is beyond the scope off the current validation that is focused on providing an initial demonstration of assay validity based on associations with disease and age. Because our sample was selected with these goals in mind, it is not suitable for such an extension (which would require, for example, dense sampling across pregnancies). Chimpanzees do not undergo any form seasonal breeding or bias in the timing of reproduction, therefore reproductive state is unlikely to have confounded the comparisons in the current analyses. Nevertheless, it is very much our plan for the future to more densely sample females during different reproductive states (e.g. pregnancy and lactation) to better evaluate the effect of reproductive effort on oxidative status. We have included a statement concerning our limited scope and future plans in Limitations revised lines 689-696. We also note the potential influence of reproductive effort when discussing age-related changes in TAC among past-prime females in Discussion lines 634-636.

I also believe that the grouping of samples by time-period instead of by individual health status is potentially problematic. Is more refined data on which individuals were sick and when available? The health of this study population seems to be closely recorded, and if this data is available, the authors should compare sick individuals with healthy individuals, instead of grouping samples by time period without knowing individual health status. The before and after periods are quite long, and it seems reasonable that individuals that experienced illness during the epidemic phase could have also experienced a separate illness during the before or after phase, potentially muddying the comparisons.

AUTHORS: We agree with the reviewer that it would be desirable if biomarkers and disease state could be more closely paired within individuals for the epidemic. While the health of this population is carefully recorded, we have some constraints in our ability to directly match samples with information on illness. First, respiratory signs are one of the few external health indicators we can observe (diarrhea, for example, is unusual and often related to particular foods), so our ability to define who is ‘healthy’ in the control periods is limited. Second, due to the nature of chimpanzees, who form small subgroups spread over a home range of >20 sq km, it is not possible to observe every chimpanzee at close range every day. This means that while we are confident in the presence of respiratory signs, we can be less sure about their absence, particularly within short time windows (Emery Thompson et al. 2018 Roy Soc Open Sci). Third, we are not sure how long before or after individuals exhibit respiratory signs they may be expected to show increases in the OS markers – indeed, this is almost certainly going to vary by marker. Finally, given the severe nature of this epidemic in chimpanzees, the acute phase associated with coughing may not be the only source of oxidative damage. Rhinovirus infections often cause scarring of lung tissue that leads to asthmatic-type coughing and inflammation after the actual infectious event (Steinke & Borish, 2016; Pediatric Allergy and Immunology; Scully et al. 2018 Emerging Infectious Disease). The elevated levels of oxidative damage (8-OHdG) and inflammation (neopterin) in individuals after the epidemic aligns with this rhinovirus biology known in humans. These considerations make it intractable for us to flag individual samples with the health status of the animal. 

On the other hand, we selected this respiratory outbreak for study specifically because its effects were so widespread and because both the onset and the drop-off in respiratory signs was sudden and dramatic. This provided a clear natural experiment. As noted in the manuscript, 87% of the chimpanzees were directly observed with respiratory signs. Given that rate and the close physical contacts that chimpanzee engage in, it is a reasonable certainty that all chimpanzees were exposed to the virus, and likely that all individuals were infected to some degree, regardless of whether they exhibited external signs. This is typical of past respiratory outbreaks recorded across chimpanzee populations: morbidity rates are often nearly 100%. This is even more likely because the period of illness was prolonged. This outbreak exhibited three peaks, when at least 50% of individuals were observed with signs in the same week. Thus, most individuals exhibited signs at several stages across the outbreak period. They may either have experienced prolonged illness or multiple cycles of infection, or both. At least one individual is known to have died from secondary bacterial pneumonia. No subjects exhibited wet coughs in the period before the outbreak, and only two exhibited them at any point in the period after the outbreak, which may reflect persistence of the infection. Thus, we made as clear a distinction as could be made between periods when nearly all chimpanzees were ill and periods when few if any were known to be ill. However, it was important for sample size to be sure that our “before” and “after” period extended for a long period of time. This means that individuals were sampled enough that any transient illness we could not control for is unlikely to bias the analysis.

We note that a strength of our current analysis is that it nevertheless does make comparisons within individuals during different time periods and does not simply compare between population means.

In humans, does OS stress increase linearly with age from the time of birth? Or does it only start to increase after a certain old age? I wonder if by looking at such a wide range of ages, there may be some signals of aging that are lost if they only occur later in life. I understand this is kind of what the “Years before death” analysis is getting at, but it may be interesting to conduct a similar age and OS analysis including only samples from individuals above a certain older age.

AUTHORS: At the reviewer’s recommendation, we now include a mixed-longitudinal analysis of age-related changes in biomarkers among individuals > 35 years old (i.e., past the age at which mortality rates accelerate). This past-prime analysis in the mixed-longitudinal sample nicely parallels the exclusion of a prime adult (KK) from the lead-up-to-death analysis. It included 9 individuals with 140 samples. According to the analysis, MDA-TBARS increased with age and both isoprostanes and TAC changed with age according to sex, such that females’ isoprostane and TAC concentrations increased and males’ decreased with age. We explain this analysis in Methods revised lines 340-347, its results within revised lines 477-485 and Table S7 and Figure S8, and discuss them in lines 535-540, 632-636 and Table 5.

Overall, this study offers an excellent discussion of potential urinary biomarkers of OS in non-human primates, and gives preliminary support for their utility in monitoring wild primate populations. The results regarding health status and OS markers could be made clearer by utilizing individual health status as opposed to group trends, if that data is available. Additionally, some of the language used in the MS is not well defined, and could be more easily understood with clearer definitions throughout. In sum, this study is a valuable addition to the literature on OS in primates.

Line-by-line comments:

Line 18- It seems as though the results of this study do not support OS as playing a central role in aging in chimpanzees, so this is an interesting choice of first sentence.

AUTHORS: We have changed “central” to “marked” and added further nuance to evidence for oxidative theories of aging and life history in Introduction revised line 16. Oxidative damage and protection are indeed key players in cellular and organismal senescence and chronic disease, however evidence from naturalistic studies highlight the dynamism of these relationships and diminish a simplistic theory that damage � aging.

Line 38- Please briefly explain what is meant by “oxygen’s reactive species”

AUTHORS: The first line of the introduction now species that reactive oxygen species are “byproducts of oxygen use”.

Line 43- “thus” implies that the second half of the sentence is a logical conclusion of what precedes it, and I don’t think this is the case for this statement.

AUTHORS: We have added that proximate damage accumulates "with time", providing the link between the functional decline that accompanies cumulative oxidative damage and aging (Line 44).

Line 46- MDA-TBARS needs to be defined before using its acronym.

AUTHORS: Done.

Line 89- Please explain what is meant by ‘redox status’. It is unclear if this is the same thing as OS.

AUTHORS: We now clarify that redox status is the balance of between oxidants, antioxidants, and repair mechanisms (revised lines 99-100). In the first paragraph of the Introduction, we explain that oxidative stress is a state of imbalance in favor of oxidants.

Line 114- This explanation makes it sound like 8-OHdG is actually a signal of DNA repair and not DNA degradation. Are these two processes always directly proportional?

AUTHORS: Yes, 8-OHdG can only be measured if damaged guanine nucleosides are excised via DNA repair. While the assay is widely used under the assumption that theses process are directly proportional, we highlight that this assumption is not always realistic. For example, DNA repair is seen to slow with age in both humans and mice (Lines 679-680). We also emphasize in Limitations that, unfortunately, there are currently no urinary assays to independently measure either DNA damage or processes of its repair (Lines 697-699).

Line 120- The author previously mentions the oxidation of lipid as an OS marker, but now refers to it as lipid peroxidation. Are these the same thing?

AUTHORS: Yes. When lipids are oxidized they produce peroxides and the production of peroxides is a process known as peroxidation. We have made it clear now that these terms are equivalent (line 130).

Line 124- Are these other physiological processes known? It would be useful to know what other processes are associated with MDA excretion in order to fully understand the results.

AUTHORS: We have added 2 such processes, namely dietary MDA content and blood oxygen tension.

Line 141- What is meant by controlled substances?

AUTHORS: We have replaced the term “substances” with “reactant and reagent” for clarity.

Line 144- What is meant by inducability of expression? I am not familiar with that terminology.

AUTHORS: We have replaced the term “inducibility of expression” with “expression in response to external stimuli”.

Line 175- You define MDA earlier, but “TBARS” has still not been defined.

Line 235- This full definition of MDA-TBARS comes very late in the MS.

AUTHORS: We have added the full definition of MDA-TBARS at its first mention.

Line 241- After visually inspecting the data “for” inflated biomarker values.

AUTHORS: Corrected.

Line 342- Why not include time of day as a covariate instead of using residuals as data? Although it is common practice, using residuals as data is thought to lead to biased parameter estimates. If you must, please provide strong reasoning for doing so. Was there considerable agreement across the population for the population slope? See:

Freckleton, R. P. The seven deadly sins of comparative analysis. J. Evol. Biol. 22, 1367–1375 (2009).

AUTHORS: Time of day was not used as a covariate in the final models because multiple samples taken from an individual on the same day required averaging to minimize psuedoreplication. We had enough same-day samples to warrant averaging, but not enough to warrant “date” as an additional random intercept. As seen in the average ß and SE for each biomarker in Table S4, the spread of slopes among individuals for markers that significantly varied with time of day was not large. More importantly, the magnitude of the slope itself was not large, meaning that biomarker values were moderately adjusted for time of day. 

Line 356- What is the response variable in this model? The composition of each model needs to communicated in a much clearer way, such as numbering them or writing them out in a table.

AUTHORS: We added a table to more clearly communicate our statistical model structure (Table 1).

Line 411- This is interesting because a recent study did not observe rises in uNEO in response to surgical trauma. Please see and discuss this discrepancy:

Higham, J. P., Stahl-Hennig, C., & Heistermann, M. (2020). Urinary suPAR: a non-invasive biomarker of infection and tissue inflammation for use in studies of large free-ranging mammals. Royal Society open science, 7(2), 191825.

AUTHORS: Thank you for pointing this out. We incorporate and discuss the findings of this study now in our Discussion (Lines 609-611).

Line 414- “One individual”

AUTHORS: Corrected.

Line 607- What does RMR stand for?

AUTHORS: We have replaced this abbreviation with “resting metabolic rate”.

Line 644- Is WEIRD an acronym here or are you just saying that we are weird?

AUTHORS: We have clarified the significance of the acronym parenthetically alongside its source reference.

Line 1375- Reference all capitalized

AUTHORS: Corrected.

6. PLOS authors have the option to publish the peer review history of their article (what does this mean?). If published, this will include your full peer review and any attached files.

Do you want your identity to be public for this peer review? For information about this choice, including consent withdrawal, please see our Privacy Policy.

Reviewer #1: No

Reviewer #2: No

Reviewer #3: Yes: Rachel Petersen

---

## [Editor Report · Decision Letter 1]

10 Aug 2020

Urinary markers of oxidative stress respond to infection and late-life in wild chimpanzees

PONE-D-20-04637R1

Dear Dr. Thompson González,

We’re pleased to inform you that your manuscript has been judged scientifically suitable for publication and will be formally accepted for publication once it meets all outstanding technical requirements.

Kind regards,

Emmanuel Serrano, PhD

Academic Editor

PLOS ONE
---

## [Editor Report · Acceptance letter]

17 Aug 2020

PONE-D-20-04637R1 

Urinary markers of oxidative stress respond to infection and late-life in wild chimpanzees 

Dear Dr. Thompson González:

I'm pleased to inform you that your manuscript has been deemed suitable for publication in PLOS ONE. Congratulations! Your manuscript is now with our production department. 

Kind regards, 

on behalf of

Dr. Emmanuel Serrano 

Academic Editor

PLOS ONE